# LiDAR- and AR-Based Monitoring of Evolved Building Façades upon Zoning Conflicts

**DOI:** 10.3390/s20195628

**Published:** 2020-10-01

**Authors:** Naai-Jung Shih, Yi Chen

**Affiliations:** Department of Architecture, National Taiwan University of Science and Technology, Taipei 106, Taiwan; chrisiyi.216@gmail.com

**Keywords:** markerless AR, LiDAR, old street heritage, building façade, architecture style, situated study, smartphone

## Abstract

Zoning conflicts have transformed Old Street fabrics in terms of architectural style and construction material in Lukang, Taiwan. This transformation should be assessed as a contribution to digital cultural sustainability. The objective of this study was to compare the evolved façades resultant from the changes made by the development of architectural history and urban planning. A combination of 3D scan technology and a smartphone augmented reality (AR) app, Augment^®^, was applied to the situated comparison with direct interaction on-site. The AR application compared 20 façades in the laboratory and 18 façades in four different sites using a flexible interface. The comparisons identified the correlation of evolved façades in real sites in terms of building volumes and components, pedestrian arcades on store fronts, and previous installations. The situated comparisons were facilitated in a field study with real-time adjustments to 3D models and analyses of correlations across details and components. The application of AR was demonstrated to be effective in reinstalling scenes and differentiating diversified compositions of vocabulary in a remote site.

## 1. Introduction

The Old Street in Lukang is replete with historical buildings (Figure 1). Lukang had originally been reconstructed from a harbor before it assumed its current fabric. Its historical identity was reshaped from a place that could hardly achieve a thorough view of the sky, to a representative icon of cultural sustainability. Its tangible pattern of development can be traced back over 300 years. The fabric, which was created from the beginning to the current setting of support, was reshaped in an evolved manner. The long-term developed urban fabric has transformed the township with the landscape of the streets being reconstructed and conserved. This evolution, which occurred within each block, dramatically impacted the construction of building style.

This transformation should be assessed as a fulfillment of cultural sustainability. The assessment of heritage can benefit from rapid technological advancements in such fields as human computer interaction, information modeling, construction, situated learning, tourism, and 3D data acquisition. For example, augmented reality (AR) has been widely and successfully applied in recent years. AR enables real and virtual information in an actual environment to be combined, registered, and interacted upon in real-time [1,2,3]. This interaction is particularly important in tacit knowledge, which is typically challenging to articulate to others [4]. Similar to case studies of environments prior to architecture design, AR can present virtual content with the real world around the user [5]. This virtual content can be created from models originated from the physical world with a high level of realistic appearance. The integration of three different modeling methodologies (TLS, GPR, 3D modelling) were developed to create an educational app [6]. Users can explore reality with new layers of information using mobile AR applications for a novel interactive and highly dynamic experience [7,8].

AR applications have been successfully implemented in a broad range of disparate fields, including navigation, education, industry, medical practice, and landscape architecture [9,10,11,12,13]. Technically, huge sets of input data have been registered to reconstruct accurate 3D city models for AR applications [14]. Moreover, extended reality and informative models have been created for architectural heritage from scans to building information modeling (BIM), virtual reality (VR), and AR [15]. Heritage building information modeling (HBIM) embodies complexity in surveying and preservation [16]. Geological view of relics and landscapes enhances the social awareness of culture and the protection of heritage [17].

Construction-specific studies were made that focused on safety with the finer details of VR/AR environment to attract significant attention [18], to simulate the environmental context and spatio-temporal constraints of various processes [19], to conduct structural analysis to determine behavior under different loading conditions of interactive 3D models [20], to setup tasks on a variety of machine tools based on target markers [21], and to contribute to online education for management and engineering [22]. Dimensional inconsistencies of building design among participants in VR seem to be partially alleviated through AR by adding a real object that is accurately scaled [23]. 

AR offers practical advantages because it allows educators to use real objects. Education-related studies were carried out focusing on STEM [24], m-learning (mobile learning), and traditional e-learning (electronic learning) [25], to enable students to locate any object represented in a given space [26], to achieve a better understanding of cultural aspects [27], to enhance laboratory learning environments [28], and to develop new learning paradigms in outdoor co-space and implement a mobile AR system for creative design courses [29]. 3D modules can be difficult to obtain, and their operation should not be too complicated in use in situated learning [30]. The link between real world objects and digital media should bring a new and beneficial quality for learning [31].

VR and AR have been of great interest to tourism researchers [32,33]. Tourism experiences can be transformed by these technologies into virtual tours and augmented tourism experiences [34]. Concerning tourism, AR can supply tourists with detailed and visceral local information [35,36,37,38]. In addition, the utilities of VR and AR in communication and engagement can be understood or improved through interactions with traditional 3D scenes or entities. To achieve this, the information model should be extended to a direct manipulation of 3D building façades which possess a strong relation to the past development of local urban architecture. The exemplification of components can not only make tourists knowledgeable in rich and novel ways, but also create a pragmatic pedagogical environment through AR technology. To promote similar experiences in a wider range of settings, a representation of past virtual environments can enrich the structure of systems for more vivid involvement either on the same streets or in remote cities. 

Problems, however, can occur with AR under different technology limits. Marker-based AR and markerless AR constitute the two major AR types [39]. Marker-based AR in outdoor settings is usually constrained by lighting conditions of the environment, weather, or time of day. To overcome this challenge, markerless outdoor tracking was developed, which uses images created from different daylight conditions [40]. Difficulty and aesthetic concerns were involved in arranging and pasting markers on building surfaces, and markerless AR constitutes a more suitable approach for diversified scenes [41,42]. 

A detailed visual representation was proven to be requisite to achieve a connection to reality to increase comprehension. Light detection and ranging (LiDAR) has been applied to 3D scan technology to document as-built scenes for construction monitoring [43,44,45] or to retrieve geometric features from a distance with a combination of aerial images [46,47,48,49,50,51]. In contrast to creating 3D models from the beginning, 3D laser scanning is an effective tool for acquiring important geometric attributes, such as detailed trees and vegetation [52]. 3D data acquisition should be extended to provide a higher fidelity experience of the particular subject. Ground scans have been shown to be feasible for accurate street data retrieval. The combination of LiDAR and AR was proven to be both interesting and promising [53,54,55], and should be explored further. 

In order to provide an effective situated comparison experience, a smartphone should be considered as a platform through which to interact with AR. The interface should also work with Cloud support with flexibility in actual locations or laboratory settings, so that architecture tours and extended field study experiences can be applied in diverse urban environments.

## 2. Research Goal

The objective of this study was to provide a support environment to assist situated comparisons of façades upon zoning conflict and evolution. The evolved façade needs to be assessed with direct interaction to correlate components and details in reality or in real sites. The comparison and interaction can be extended from the study conducted at Lukang Township to a remote site as a way to present and verify research findings. 

This study monitored and mapped the profiles of building forms upon zoning changes made within the same street block in Old Street. Urban zoning control was commonly applied to existing open spaces, such as streets, to separate commercial areas and form preservation areas. Since 1986, street blocks near Old Street have been controlled by creating separation of the two types of areas within the same block. 3D data have to be documented as needed in order to obtain an instantaneous side-by-side study of the effects of the two types of changes to the same street block. It is noted that this process can then also be beneficially applied to diverse adaptions of form in various other cities. In order to make a comparison to a real scene, a higher fidelity of appearance must be provided. LiDAR and AR-based building form mapping and monitoring are needed to achieve this. In some cases, a mesh model must also be applied to overlap with the wireframe appearance of data for clear indications of molding locations.

## 3. Research Methodology

A situated comparison was carried out in this research. In order to verify the evolved façade, the proposed process included inspecting restructured space, applying 3D scans and supporting interaction devices, correlating both sides of the façade in AR, AR-assisted comparisons and scene reinstallations, and verification. The comparison needs an accurate assessment of configuration and a user-friendly interface with which to work. The assessment should be made by correctly reconstructed spaces or objects as a reference. In order to match with a realistic scene in an architecture site, reconstruction by 3D scans offers an effective approach to capture precise dimensions and correct relative locations in as-built form. The realistic 3D scan data will facilitate field visits and comparisons with great visual similarity and dimensional accuracy, instead of a simplified representation of vector drawings of the original design. The comparison also requires a user-friendly interface to correlate subjects in the foreground and the background in real time. Indeed, the AR setup establishes an optimal environment to interact with photorealistic subjects and backgrounds.

We aim to apply a public platform to support these zoning conflict comparisons. Our former researches developed a customized AR app using ARkit 1.5^®^ to decomposed specific spatial structures with display options [41]. Although the research purposes were fulfilled, the recycling of the original codes or design was not optimal. A platform which provides everyday Internet 3D browsing experience for the young generation would constitute a promising choice. A commercial platform, Augment^®^ [56], which provides end-to-end AR, was selected as an approach for 3D visualization and communication in this paper. 

In contrast to pre-defined solutions of e-commerce, sales and marketing and platform design, we intend to utilize it as a study environment for architectural façades by comparing its development due to zoning changes and geographic differences. This environment should support façade overlay with/out the real context of background accessed either in the same site or in different cities. An easy access without a predefined structure may provide another type of flexibility in architectural interaction, with the least possible development and maintenance costs.

An emphasis is given to the preparation of 3D façade models. A similar concept of scalable model can also be found in Sketchfab^®^, which takes the 3D point cloud for the VR display. Augment^®^ was selected because multiple AR objects can be displayed in the same scene, to be interacted with, within a single subscriber account. Although the display of each individual object cannot be turned on or off as needed, the application paradigm can be found in interior design with multiple pieces of furniture involved in a layout process.

The proposed process was made from restructuring, correlating, 3D data retrieval and conversion, scene comparison or reinstallation, and finally verification (Figure 2). The flowcharts include, (1) the stepwise process of situated comparison by issues, subjects, and locations; and (2) the development of data and related conversion to support the study.

## 4. The Restructuring Background of Old Street 

Chenggong Road, which was the location of the former Lukang Creek (Figure 3), was originally a trade distribution center. Cargo, which was unloaded and sold on the riverside, was rearranged and repackaged after being moved indoors and sold at store fronts on the other end of buildings facing Putou Street and Yaolin Street. The entire process occurred in a house approximately 4 m wide and as much as 40 m deep. The houses located to the west side of Old Street possessed a harbor-facing geographical advantage, and the houses located to the east side were generally more inconvenient for the commercial process. An owner had to bypass Old Street to reach Houche Lane, where the service entrance was located. Houche Lane is located 40 m to the east of Old Street, and faces the rear entrance of buildings on Zhongshan Road.

Lukang Old Streets have had a well-preserved Minnan architecture since the Quin Dynasty. The region defined by the green boundary (Figure 3, middle) is a heritage preservation area, while the commercial area is in the red boundary. The famous Old Street is located at the center of the preservation area. In general, the street and houses on it exist under the same zoning classification. The two areas, however, are now divided by a jagged form that follows the property line. The consistency of the façade in each area stopped in the middle of a block. The lot, which was previously an undivided property, was separated irrespective of the original space hierarchy at which approximately the third or fourth hall was located (Figure 4). The traditional layout between the main and rear entrance was no longer connected from the interiors. A new entrance had to be created, or the interiors had to be remodeled from a space next to the service entrance, in order to establish a feasible living space. This type of differentiation was achieved by a long-term development of urban fabric (Figure 4, right). The restructuring of living spaces has occurred since 1986. Since Houche Lane is located next to both the preservation and commercial area, it is necessary to explore how the building style has developed. 

Due to the spatial structure arrangement used to connect the harbor front to the shop front, the property line is linear, with a large depth-to-width ratio. An old property usually occupied an entire street block from one end to the other. Long-term zoning development had reconstructed two different zones within a single street block, and dramatically reshaped the urban fabric in a 3D configuration. The division separated an integrated traditional hierarchy with sufficient space from 3 to 4 fully developed halls to 1–2 halls with insufficient space for either part of spatial structure. Moreover, each space type was developed with different materials, vocabularies, styles, and spatial structures within the same block. In fact, great contrast was established by having a horizontally-developed structure of traditional architecture next to a vertically-developed structure of street-facing buildings.

The zoning differentiation created problems for the truncated part of the traditional architecture. Specifically, the dated layout and differentiation of original functions in various zones constrained the size and flexibility of existing spaces from supporting many types of functions. In order to meet commercial and daily needs, and simultaneously maintain the environment, second building skins were created in an additive or subtractive manner (Figure 5). The skin was accomplished by extending the missing part of an interior space to the exterior walls or open spaces for laundry facilities, HVAC (heating, ventilation, and air conditioning) units, canopies, posters, or shop signage, in fixed or flexible settings. Fixed canopies, for example, were usually installed with an associated platform for booths to sell merchandise. All of the commercial installations indicate a strong purpose of interstitial space between the street and indoors. 

Additive interstitial space was developed with an object-subtraction approach of posters or free-standing antennas. Consciousness and responsibility were also developed to rearrange the layout of booths, landscape, and personal belongings. The addition or subtraction of installations represents an adjustment to the balance between cultural identity and facilities for commercial activities within the same historical street space. Indeed, the street space was constructed upon a hieratical structure of demands and supporting facilities for 300 years. 

## 5. Correlating Both Sides of the Façade 

Since zoning led to the development of different evolving façades and building vocabularies, a comparison made across zones, or even across cities, should be able to confirm deviations and achieve a broader understanding of the architectural transformation. The first problem encountered was determining how to bring the façades together from two opposite sides of a single street block. Although two photographs or drawings of elevations can be placed side-by-side or even overlapped to highlight changes in an off-site laboratory setting, 2D-based verification is made with a limited understanding of depth. Otherwise, sections have to be brought up as needed in order to correlate the differences.

To solve the problem of correlating 3D data in different locations, we proposed a solution for off-site, or even on-site, inspection. The approach applies the AR interface to bring two 3D models together for an intuitive inspection with a beneficial manipulation of transition, scale, and/or orientation. In contrast to superimposition of real scenes using a virtual 3D model created by software from the beginning, the fidelity of the virtual model should provide sufficient information to identify dissimilarities in visual and structural details. 

A total of 40 buildings, i.e., 20 on Old Street and 20 on Houche Lane, were selected according to four categories based on the types of changes existing on both sides (Figure 6). Approximately 63% of new constructions on Houche Lane are taller than the buildings that used to be located within the same property boundary, while another 15% of places have remained vacant. 

## 6. Design of the Situated Comparison in AR

The application of AR was carried out based on a combination of appropriate types of 3D models and background scenes in a remote site. Since the models can be downloaded as required, the comparison becomes portable, in terms of the database, with the support of a smartphone platform. 

The AR application serves the following three purposes:Correlating parts at different locations: Geographically-distributed building parts were brought together side-by-side or overlaid between different contexts to achieve a paired comparison.Applying higher fidelity of part appearance: The selected level of fidelity can be a critical factor in the comparison. The target façades on Old Street were documented using a mid-range 3D laser scanner for visual and structural details in higher fidelity.Situated comparison using a flexible transformation interface: Transition, scaling, and/or rotation were made possible by adjusting parts for alignment in terms of proportion, length, width, or depth.

### 6.1. A Reality-to-Reality Environment and a Wireframe-to-Reality Environment

In order to fulfill the needs of an accurate comparison, the application facilitated an appropriate manipulation interface and versatile use environment in the following ways:A reality-to-reality environment: Two mesh models were located side-by-side to facilitate the inspection of alignment between moldings or openings across the two models. An overlaid layout was selected to support the inspection of depth in various levels.
▪A 3D mesh model over a real scene: A background context-concerned environment was applied for a meaningful study of development.▪Two 3D models on a plain, simplified, or context-free background: Two 3D mesh models were applied simultaneously with or without the real scene in the background.▪A number of 3D models over a real background: This is presented in a way that is similar to a set of instances to be applied for inspection.A wireframe-to-reality environment: Although vector drawings usually have lower fidelity of details, the enhanced description of an object’s edges provides an advantage when overlaid on top of the mesh model. The wireframe constituted a set of very thin polygons that can be seen through to the 3D model behind it.
▪A wireframe model over a real background.▪Two 3D models on a plain, simplified, or context-free background: A 3D mesh model and a wireframe model were applied simultaneously with or without the real scene in the background. The latter was represented in a more abstract form for simplicity of comparison.▪A number of wireframe models and 3D models over a real background.

### 6.2. Remote Comparisons

Comparisons were usually carried out in “remote mode”, in which a 3D model was brought up or downloaded from the Cloud at a different location.

The opposite side of the same street block: The model was used to compare the different façade style developed on the opposite side of the same street block.Different street block: The model was placed on a property where the original house was demolished.Different city: The model was attached to a façade located in a different city, where the original style evolved in a dissimilar manner.In a laboratory setting: The model was downloaded in a laboratory setting without reference to any real background for a background-free context test.

## 7. 3D Scans and Supporting Interaction Device

In contrast to drawing-based comparisons made by attaining graphics to illustrate differences, the assessment should be made by higher fidelity 3D digital data which support a real-scene-based on-site or remote inspection. Moreover, in order to support the on-site study, an accessible device should be provided with the support of a 3D display and interaction, in addition to access to Cloud data. Since AR has been broadly applied in architecture for a combination of 3D models and real scenes, the assessment of differences can be achieved in an AR platform if a high level of interaction, higher fidelity of the model, and domain-specific transformation are supported by a smartphone platform using iOS^®^ or Android^®^.

A street 200 m long was scanned (Figure 7). A mid-range Faro Focus 3D S120^®^ laser scanner was applied. In total, 4.97 GB of point cloud data was retrieved and stored in PLY format. The scan of the as-built street scenes obtained a high fidelity of street façades. The “Poisson surface reconstruction” of the mesh model was created from the point cloud by selecting the “output density as SF” option in CloudCompare^®^ (Figure 8, left). The mesh creation process was made in Geomagic Studio^®^ (Figure 2). A “wireframe-like” mesh model (Figure 3, right) was also created based on a field survey using AutoDesk 3DS MAX^®^ (Figure 8, right).

A collection of 20 paired façades on opposite sides of the same street block (Figure 9, left) was created by converting the 3D scan data to the mesh models from CloudCompare^®^, Geomagic Studio^®^, AutoDesk 3DS MAX^®^, or Meshlab^®^. The most frequent connection to the models was through the smartphone app. An iPhone 10^®^ with iOS 12^®^ and a Realme X50 5G^®^ with Android 10^®^ were utilized as the smartphones for field comparison with Cloud links available from Augment^®^ (Figure 9, right). A subscription Augment^®^ app was applied as a ready-made general-purpose platform to alleviate AR software development and maintenance effort. It is an e-commerce platform for sales, marketing solutions, and design solutions. In this paper, we use it as an “end-to-end AR and 3D visualization and communication platform” [56] of cultural artifacts with so-called “online shopping experience” replaced by “online situated comparison experience”.

## 8. AR-Assisted Comparison and Scene Reinstallation

The comparison was conducted in the following four stages: (1) selection of model pairs; (2) rough alignment; (3) detail matching; and (4) context elaboration. Two façades were selected with a similar appearance. Rough alignment was applied in order to align the two models by translation or rotation, and was usually made to the corner of the façade. Detail matching applied translation and proportional scaling to a certain dimension in order to show the difference in height or width of a specific component. Adjustments had to be made repetitively by selecting appropriate anchor nodes or edges as a reference. This is required because, for example, two windows with the same width do not necessarily have the same height. The first three stages enhanced comprehension of the aspect of focus, increased user familiarity, and achieved the desired manipulation. The final stage encouraged users to inspect any façade that they chose in the AR app by overlaying the selected model on-site with the same stages to elucidate the transformations. The stages are intuitively performed with a few taps and movements of one or two fingers.

Situated comparisons in different cities were made through model adjustments, detail checks, and corresponding components (Figure 10). The initial adjustments allocated a façade model in place by referring to a number of heuristics. The reference informed the researcher of major differences in proportion. More detailed checks were made to the dimensions, partitions, or materials. Adjustments frequently occurred in switching between the reference of ground level or wall molding, as a way to elucidate the difference in composition, proportion, or missing components.

The comparisons were conducted at five sites: one laboratory site; one site in Lukang; and three sites in Taipei. AR made a great contribution in comparing the façade styles on both ends of a block. The façades on Lukang Old Street facilitated architectural study conducted from different viewpoints of historical culture, building styles, and scales. A stepwise operation and third-person photographs concerning on-site operations are illustrated in Figure 11. The process was performed in several steps from scanning a QR code, pointing a smartphone camera and moving around to determine the ground plane, tapping the screen to anchor the 3D model initially, and scaling or moving the model in reference to the background. Each comparison or visit was followed by discussions which were separated into three parts: (1) the preliminary presentation of models; (2) the on-site physical adjustments; and (3) the post-visiting group-discussion of findings with screen shots for verification. The last part was helpful to recall ignored components or missed mentions of adjustments. While most of the field work was conducted by an individual researcher, some of the field work was also performed by two researchers, which enabled them to compare their findings.

### 8.1. Laboratory Site

The first set of comparisons, which included the original 40 models with two sets of heterogeneous façades, were made to the set of wireframe-like models (20 models) and mesh models (20 models) on a plain background in the interiors. Findings were obtained during the interaction between the two types of models. Even without the real context of a scene presented in the background, different forms and layout compositions were identified in the two sets. The laboratory site was used to make a comparison irrespective of background (Figure 12) or prior to the field application. Using a mesh model and a wireframe-like mesh model highlighted each one’s geometric characteristics.

### 8.2. Lukang Site

The second set of comparisons was made in Lukang by the same researcher with mesh models on real scenes under a homogeneous appearance. The former indoor and off-site comparison results were verified in real sites with further emphases on coherence of neighborhood heritage relics, continuity of alley fabrics, reinstallation of facades in demolished sites, and the evolved form of pedestrian arcades. The four locations in Lukang were selected according to condition, if it was (1) empty; (2) demolished; (3) in current form; or (4) located next to a historical artifact. The issues involved consistency of style or continuity in terms of building components and construction methods. An urban conversation was restored to the past conditions.

The façade on Yaolin Street was relocated to Houche Lane for a more straightforward side-by-side comparison (Figure 13, from top to bottom, left to right):The first comparison was conducted on buildings with a similar scale. For example, its eave is located at approximately the same height as that of the next building. The AR simulation of a coherent building component created a scene of continuity which was rarely found in Houche Lane.The second comparison was carried out next to the Lukang Gate, which is conspicuous among the modernized constructions. The inconsistency in style was alleviated with the traditional façade installed next to it in front of an empty site. AR was used to reconstruct a similar historical street scene.The third comparison was performed to compare different building styles. The side-by-side layout clearly revealed the difference in construction materials, opening partitions, and mass volume.The fourth comparison was conducted at an empty site where a former red brick building was demolished. The site was left unused and unconstructed. The same model was inserted in order to restore continuity of the original street scene.

### 8.3. Taipei Site

Another comparison was carried out for three sites in Taipei, based on the development of pedestrian arcades (Figure 14). One comparison was made to one of the earliest street buildings, the Linwuhu House, constructed in 1851. The most significant difference is whether it is a recessed pedestrian arcade from the façade or extruded as an additional canopy outside of the façade.

The third set of comparisons was conducted based on the results of the second set by selecting five unique types of facades with different variations of second skins, booth settings on shop fronts, and relative dimensions, for a more detailed inspection through adjustments and assessments of non-matching components. Most of the previous findings were reconfirmed and elaborated through the evolved design of certain components, such as the arcades or the transitions between indoors and outdoors with/out the articulated level difference of steps. The homogeneous-based comparison was successful because the context was richer than that of the second set. The urban fabric was replete with diverse objects and activities, which were evaluated according to the types involved and confirmed by the frequency of adjustments. Four facades were compared twice, each with daytime and night scenes.

The major difference in the situated comparisons between the second and third sets was the richness of context that informed the researcher’s comprehension of the evolved façade in today’s fabric setting. Many third-person photographs and screenshots were taken to elucidate the so-called interference as part of everyday life or the cause of evolvement, without objects hidden deliberately. The unselected photos revealed the accidental involvement of pedestrians or tourists in the period of the Covid-19 pandemic.

### 8.4. Reviewing of Situated Comparison

A matrix was created afterwards to review each comparison according to the adjustments made during each field trip (Table 1). A table was constructed based on the researcher’s description and screenshots. The evaluation process was based on items defined in the flowchart. Similar adjustments, which were shared by almost all comparisons, were related to the transformation of the entire model and the alignment to ground level. A more detailed assessment, including the width and height of openings, was not performed, since the 3D models were opaque to the background. Alternative operations were carried out by moving the model sideways or vertically to reveal the details hidden behind. This problem was subsequently solved by providing semi-transparent models to elucidate the missed details in the background during the AR simulation.

## 9. Research Findings

The application of AR assisted the differentiation of diversified vocabulary composition or the reinstallation of scene in a remote site.

### 9.1. Volumes and Building Components

3D models were created for comparison. As shown on the top of Figure 12, the 3D polygon model is a typical Minnan street house on Old Street. The wireframe represents a new concrete building on Houche Lane after the previous house was demolished. The latter, which is located in the middle of Old Street and Zhongshan Road, was influenced by the vocabularies of the two streets alternately for the past 300 years. For example, a concrete guard railing on the second floor can be found on Zhongshan Road. However, new wood door panels and vintage components also emerged on Old Street. The zoning development in the middle of the block had razed the old courtyard. The former, which adopted an old symmetric layout of openings, also ended up with a very limited area of windows. In contrast, the irregular opening layout of new concrete buildings had interior lighting that was far superior to that of the previous houses. The zoning policy had changed the living and environmental quality on either side, and simultaneously gradually adopted vocabularies from either street as a connection to, or recognition of, the local cultural identity.

The overlapped AR models on the bottom right of Figure 12 show that part of the wireframe model is obscured by the extruded canopy at approximately the height of the second floor. This means that the height of the ground floor on the Old Street side is approximately one half of a story higher, and is the altitude at which a canopy was installed.

The evolved space on Old Street, which is located in a conservation area, has most of the building façade covered by commercial facilities. Merchandise booths also appear on the shop fronts. In contrast, the situation is different on Houche Lane, which is located in the commercial area. While there was a former restriction of bay width of approximately 4–5 m, a rearrangement of property occurred on Houche Lane, as three lots were merged into one large apartment property.

### 9.2. Pedestrian Arcades on Store Front 

The Minnan style street houses in Lukang and Dihua Street, Taipei, exhibit noticeably dissimilar pedestrian arcade designs (Figure 15, top). The arcade, which is a covered passage corridor at the front of a building, can also offer utility as a weather shelter. The arcade on Dihua Street was designed to be set back from the projected property boundary, in contrast to the canopy-like shelter outside of the property boundary in Lukang, as the street does not open to the sky. Although the latter was reconstructed into an environmentally-friendly open-air design, resemblance was expressed by the vocabulary and component composition applied to the façade (Figure 15, bottom). In addition to the physical layout, the arrangement and the words of spring couplets for Chinese New Year, located on the top and both sides of the main entrance, present the business identity of the owner. A similar connection between the architectural components and tradition custom was also established by where the couplets were posted.

Approximately 85% of the Lukang population came from Quanzhou, China, during the period of Japanese occupation [58]. The recessed design of the arcade, which was not applied in Quanzhou, was also not initially implemented by early immigrants in Lukang. The interstitial space, defined by the arcade or canopy, significantly improved convenience for both owners and customers. Although the mechanically-retractable rolling canopies reshaped the appearance of Old Street, the components were installed without violating the original layout. Cultural sustainability was maintained to a certain extent, and living demands were met by using a flexible extension to the second skin.

### 9.3. Differentiation of Street Façade in Early Days by Reinstallations

Due to the geographic advantage of Lukang as a harbor in its early days, both the Minnan style and the Baroque (or Western) style of street houses were preserved. The reconstruction made in the period of Japanese occupation also reshaped the street style on Dihua Street, Taipei. Specifically, the old façade in Lukang was superimposed on that of Dihua Street in AR to reinstall the street style to the form of its early days in virtual space, with the reality created by referring to the remote site. Figure 14 illustrates the different opening compositions on the old ground floor and a new construction of a three-story high structure.

This AR-based study facilitated the illustration of differences in façade vocabulary and composition, material composition, reconstruction due to property transfer, and coherence in construction materials. Due to the open design of the pedestrian arcade on the ground floor and the vertical organization of spaces, the façade on Dihua Street presents a dissimilar vocabulary and composition. Symmetry, however, remained on both sites. In addition, the façade was primarily made of wood in Lukang, compared to the red brick and cement stucco found on Dihua Street. The former has openings made of wood, while the latter features openings comprised of metal frame and glass. The old and small houses, which were allocated in individual lots, were reconstructed into apartments with a volume that significantly transformed the local fabric. Materials similar to the selections of Old Street were also applied for coherence of historical appearance. Wood openings or bricks were used even in cases in which the new buildings possessed a concrete structure.

## 10. Discussion

### 10.1. Contributions

A more direct connection to architectural study was achieved by integrating façades, 3D data, and application environments. Correlations were made to the evolved design or style of pedestrian arcades on store fronts to fulfill functional needs under geographic variances. The differences between old façades and new ones were verified by the components and their compositions. Differentiation of street façades in early days was achieved by reinstallations of façades in front of a demolished site with an emphasis on coherence and continuity to existing eaves. The elevation of façades was differentiated by how it affects the installation of canopy. On-site interactive comparisons were introduced without architectural drawings.

The reality of artifacts was enhanced in 3D data. A connection between architectural artifacts and virtual representation was obtained as an extended sustainable application of as-built data. A variety of 3D mesh model types (wireframe-like, semi-transparent) was applied for enhanced comparison results.

The study applied a common environment of Cloud access, a markerless AR, and a convenient smartphone to architecture field study. The integration enabled a situated comparison to correlate the results under geographic differences. A combinative approach of complicated scan data and popular app was applied. The general-purposed platform was feasible for situated architectural study. The application was made possible for on-site comparisons even in remote sites. The visualization and communication of 3D models was incorporated to reduce development and maintenance effort, with more emphasis placed on preparation of 3D models.

A set of models converted from as-built scenes created a more coherent appearance to the environment. Twenty (20) sets of models were created and compared in seven sites with different numbers of adjustments and findings. The situated operation involved researchers in a local environment replete with both contemporary human settings and reconstructed building styles. The operations contributed to the environmental awareness and cultural sustainability of building style.

This approach can be conducted individually or as a team at any location with the support of a Cloud database. The entire adjustment process can be recorded for post-visit review. Live broadcasting is also possible in the current pandemic situation between researchers and team members, or students and instructors. A collaborate environment should be established for architectural styles and construction components as real-time situated studies of geographic differences in the future.

### 10.2. Field Comparisons

Situated comparisons succeeded in identifying component-related variations, such as the discovery of different building bay widths, extended ceiling heights on ground floors, or level distances between indoors and outdoors. The recognition of the subject and environment was always enhanced by the surrounding urban fabric and furniture. With the extended support of media, video recording of the operating process was made possible for the entire adjustment process.

Situated comparisons, however, did not always succeed. For example, a drifting problem occurred, which may cause inconsistent viewing perspectives. Depth display was another issue which created a certain model overlaying effect over pedestrians or vehicles at near depth. Moreover, restriction of operation clearance occurred when the smartphone camera was barely wide enough to include the whole façade in a narrow alley. Ground level selection and anchoring can also be tolerance-prone in a few cases in which the ground floor and façade were not at a perfect 90 degrees. In addition, during recording site activity, a third-person picture of situated operation by researchers could be difficult in terms of balancing the brightness between the façade and smartphone screen. Furthermore, interference of pedestrians, tourists, and vehicles frequently occurred (Figure 16), and each comparison took approximately 18–30 min to complete. Lighting conditions may also change the saturation of models upon automatic adjustments performed by the phone camera. In contrast, it was relatively easy to re-situate comparisons in the laboratory or indoors.

The success rate of taking photographs for post-visit evaluation was near 90% for one-person screen shots of the model and the façade from the app. This rate, however, dropped to half when photographs were taken with the screen and the façade cross street included at the same time with a clear display of content on the screen. The rate dropped even lower when taking photographs with the screen, the researcher, and the façade included simultaneously. The SLR camera function of narrowing the aperture to cover longer focal lengths was not applicable in a smartphone.

### 10.3. Switch of AR Development Environment

Former app development required substantial time and effort, as well as periodic software development kit (SDK) updates. Customized needs led to predefined architecture and data, which may not be exchangeable among studies without redevelopment. It also did not accept all kinds of data, such as point cloud, or formats for the best representation of reality. In the present study, a different application approach emerged for an app which is relatively simple and capable of working with most kinds of data with higher visual fidelity.

For the adoption of the ready-made app, the experience of interface, Cloud access, and QR code access were consistent with similar app experiences. It is similar to a webpage tool, subscription e-commerce platform, or virtual store or classroom with which students are familiar and use frequently. This transferrable experience and familiarity of tools lowers the learning curve benchmark. The instructor was also a content provider who presented students with forms of models made by different mesh creation tools. A customized app may achieve the result more rapidly, but offers less flexibility to explore the subject in real time.

Fast development of contents from creation to application was made possible using different content development approaches. A user can select a simpler CloudCompare^®^ approach or a Geomagic Studio^®^ approach with more variable control of mesh quality. The access of contents was sharable and yet secured. It is similar to a shared AR study group that can have models contributed by a number of involved people, and discuss shared experiences through screenshots or video-conferencing.

The advantage of the scan data provider in this study is that it enables an AR environment with a realistic model for architectural situated-study environments. Instead of a customized application for a specific purpose, it is an open app that hosts a general-purpose application. The data or models that we provided were relatively convincing. As a consequence, more emphasis can be placed on the contents. While the flexibility of interaction created new findings, the transfer to a ready-developed app also enabled an accurate assessment of building styles.

### 10.4. Limitations

The study also possessed certain limitations. The model may have shadow-casted differently from as it appears on the façade, depending on the weather or the time of day. On/off display switches were also not supported individually for each object without additional scripting effort, same was the customized annotations added in mixed reality (MR). In addition, the device display was small. Finally, it could not accept a 3D point cloud model, and its conversion can be effort-intensive.

## 11. Conclusions

The study applied a common environment of Cloud access, a markerless AR, and integration with a convenient device for architecture field study. This research applied a combination approach of complicated scan data and a popular app. The application of a general-purpose app for architectural study was made possible with specific models for on-site comparisons, and further extended the study to remote sites. A set of models, which was converted from 3D scans of as-built scenes, was made with a more coherent appearance to the environment. An optimal combination of sensor data and architectural manipulation of historical space was achieved in a user-friendly AR interface. Moreover, the application of 3D as-built data is demonstrated to be sustainable, even for the purposes of knowledge tours and pedagogy of architectural history.

Future works would investigate cross-application interaction platforms that support different architectural contents and issues. AR should accept all kinds of data, including point cloud, as a more direct connection from one reality to another reality, or from different input devices to reality. An open environment should also be established for self-contained study of individuals at any location with group support of a domain-specific Cloud database.

## Figures and Tables

**Figure 1 sensors-20-05628-f001:**
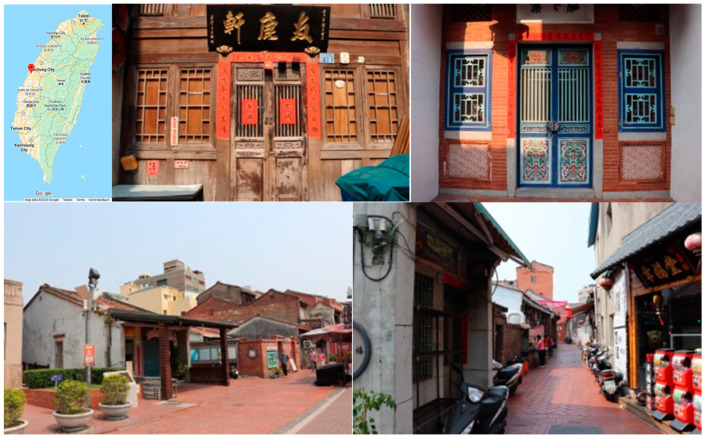
Street scenes of the Old Street in Lukang, Taiwan.

**Figure 2 sensors-20-05628-f002:**
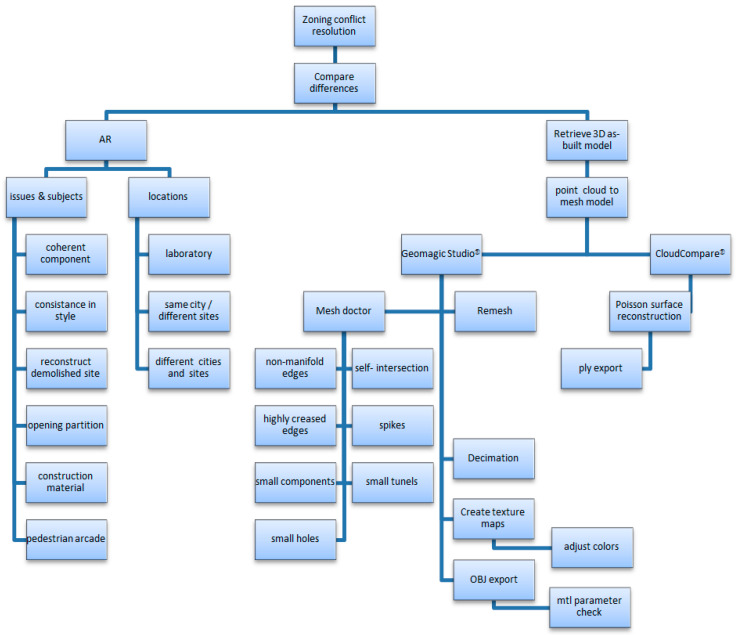
Flowchart of augmented reality (AR)-assisted situated comparison and the data conversion process to support the study.

**Figure 3 sensors-20-05628-f003:**
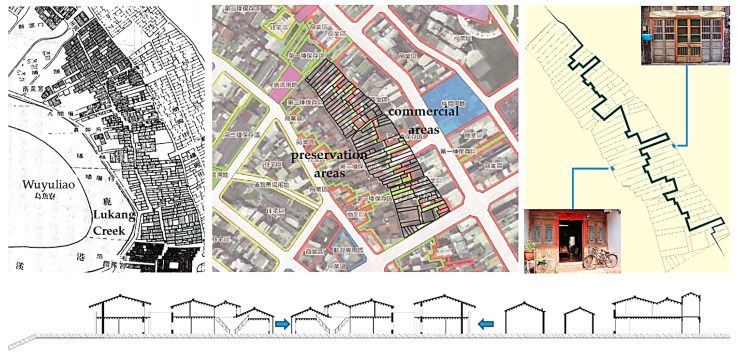
Site plan of the old harbor [57], zoning map, and differentiated street block (**top**, from **left** to **right**); cross section of four streets (**bottom**).

**Figure 4 sensors-20-05628-f004:**
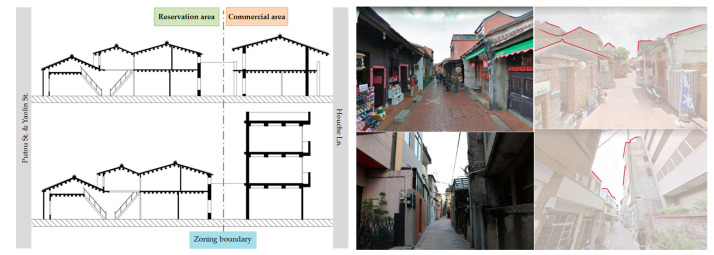
The section before and after zoning reconfiguration (**left**); the different space and the roof configuration of spaces on both ends of the block (**right**, **top**, and **bottom**).

**Figure 5 sensors-20-05628-f005:**
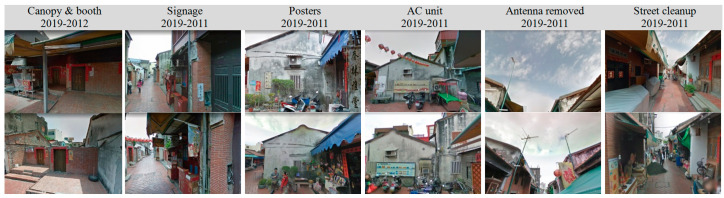
Objects added to the second skin between 2019 (**top**) and 2011 (or 2012) (**bottom**).

**Figure 6 sensors-20-05628-f006:**
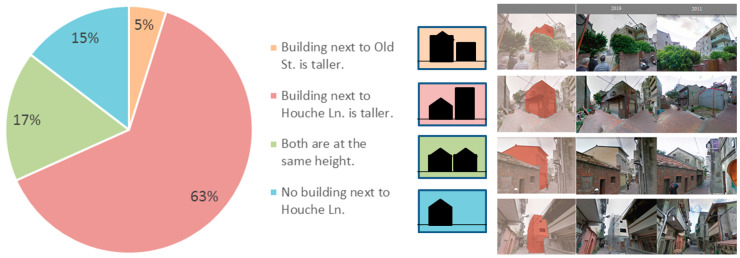
The four volumetric combinations of the constructions on Old Street and Houche Lane (**left**), and the new constructions that occurred on Houche Lane between 2019 and 2011 (**right**).

**Figure 7 sensors-20-05628-f007:**
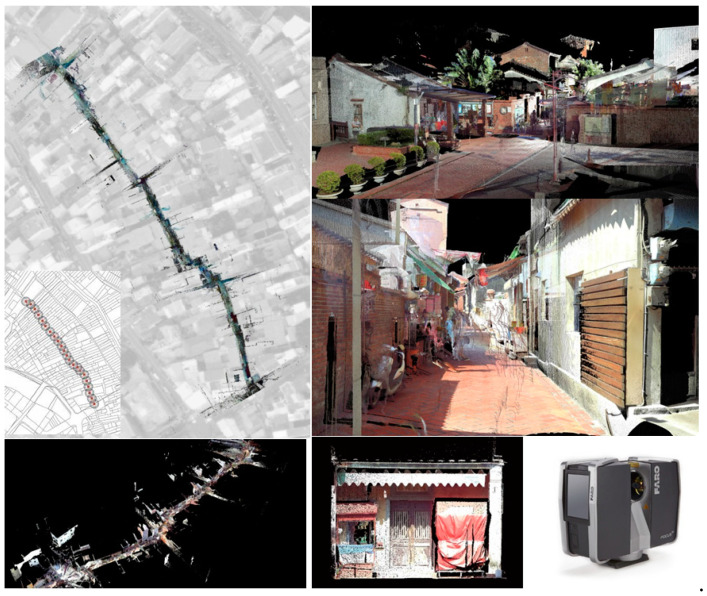
3D scans of Old Street shops (**top**, **bottom left**) and applied Faro Focus 3D S120^®^ scanner (Faro, Germany) (**bottom right**).

**Figure 8 sensors-20-05628-f008:**
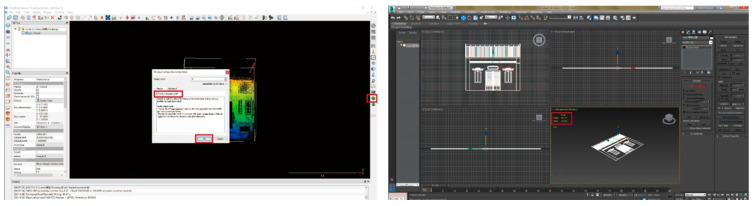
Mesh model creation process in CloudCompare v2.9.1^®^ and AutoDesk 3DS MAX 2015^®^.

**Figure 9 sensors-20-05628-f009:**
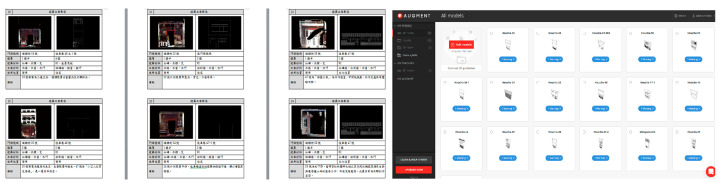
A collection of paired façades on the opposite side of the same street block (**left**) and the model list in Augment^®^ (**right**).

**Figure 10 sensors-20-05628-f010:**
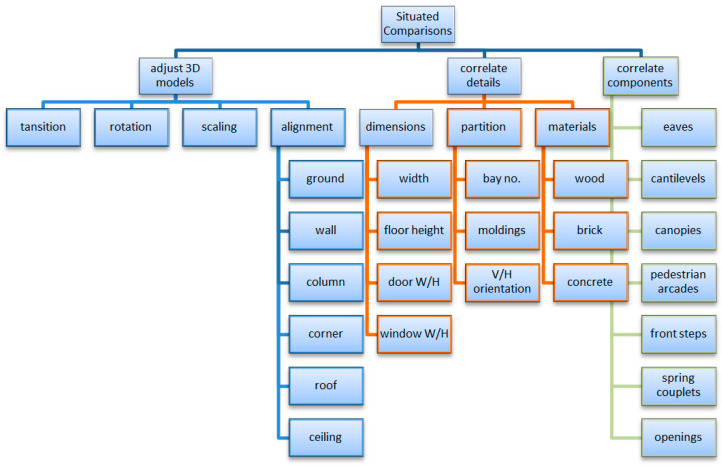
The situated comparison process.

**Figure 11 sensors-20-05628-f011:**
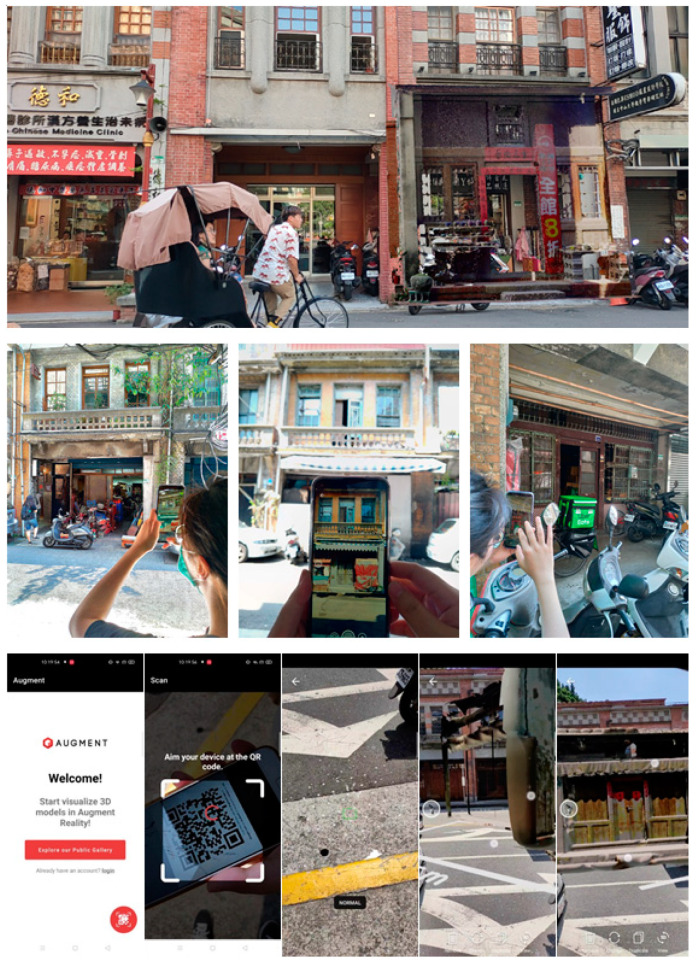
The screen shot (**top**), third-person photographs (**middle**), and the stepwise operation process: launch app, scan QR code of a selected 3D model, detect plane, tap to allocate the model, and adjust the model’s scale, location, or rotation angles (**bottom**).

**Figure 12 sensors-20-05628-f012:**
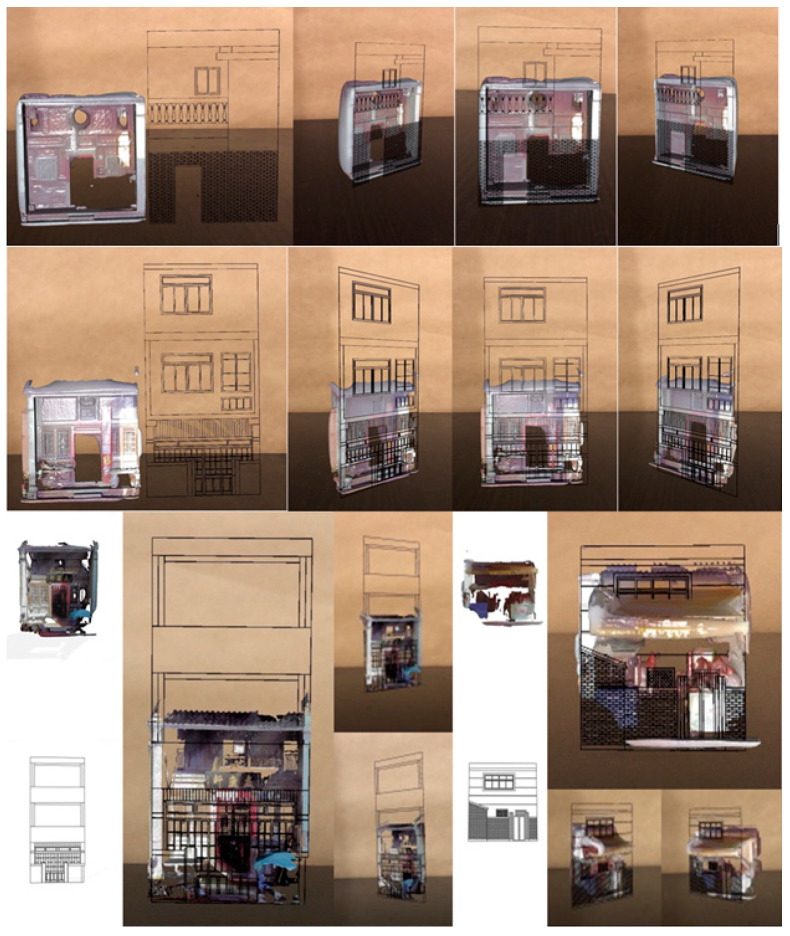
AR comparisons of the booth, material, and composition of façades on the two streets in a laboratory setting, using a mesh model and a wireframe-like mesh model. The **top**, **middle**, and two **bottom** images represent four cases of comparisons.

**Figure 13 sensors-20-05628-f013:**
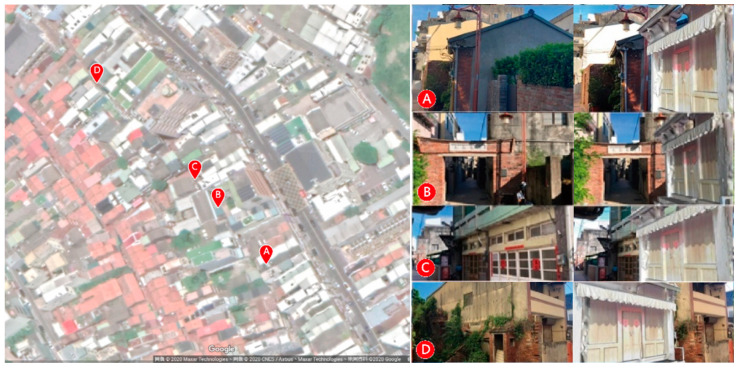
AR screenshots of relocating façades on alternative sites in Houche Lane, Lukang.

**Figure 14 sensors-20-05628-f014:**
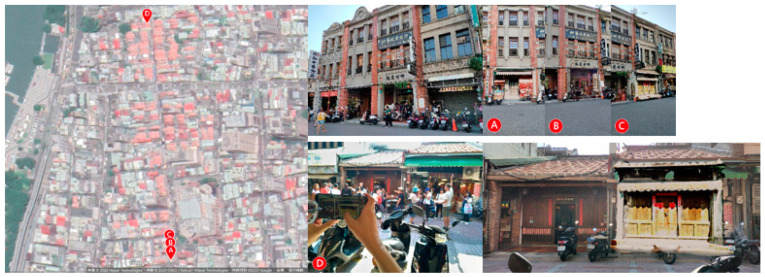
AR-based simulation and comparison of façades in Dihua Street, Chifeng Street, and Bopiliao, Taipei (from **top** to **bottom**).

**Figure 15 sensors-20-05628-f015:**
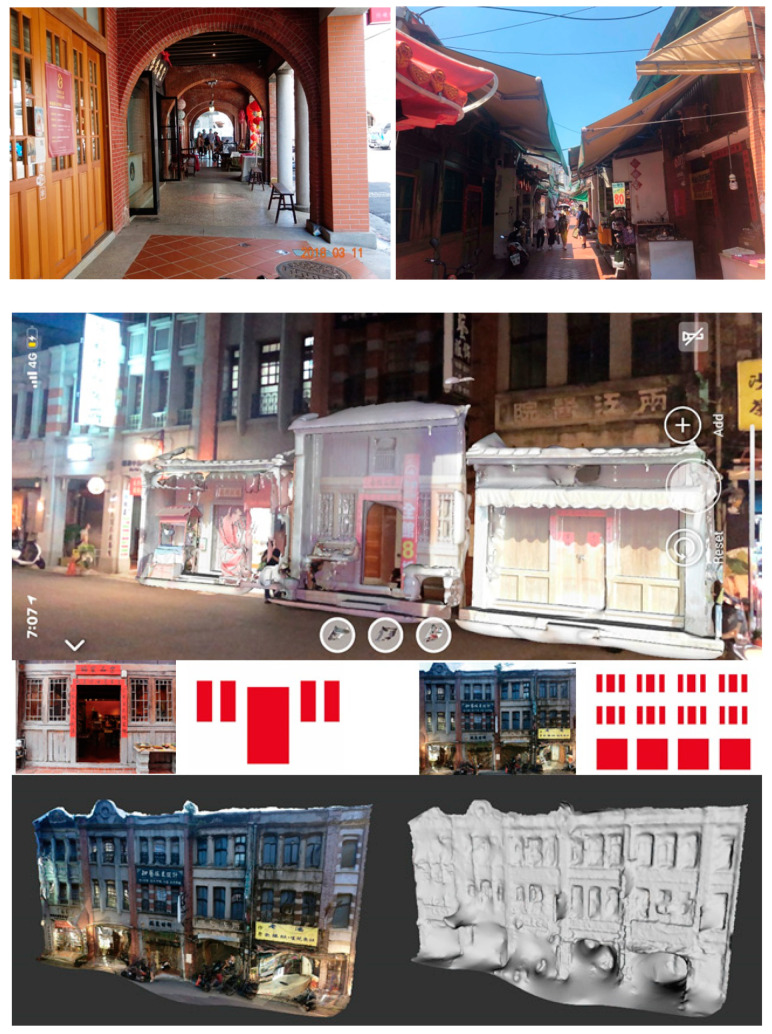
The pedestrian arcade with/out a recessed setting (**top**), three façades applied to recessed setting in Dihua Street (**upper middle**), comparison of the different opening composition on an old ground floor on Lukang Old Street and a new three-story high construction on Dihua Street at a remote site (**lower middle**), and the original 3D façades model of Dihua Street (**bottom**).

**Figure 16 sensors-20-05628-f016:**
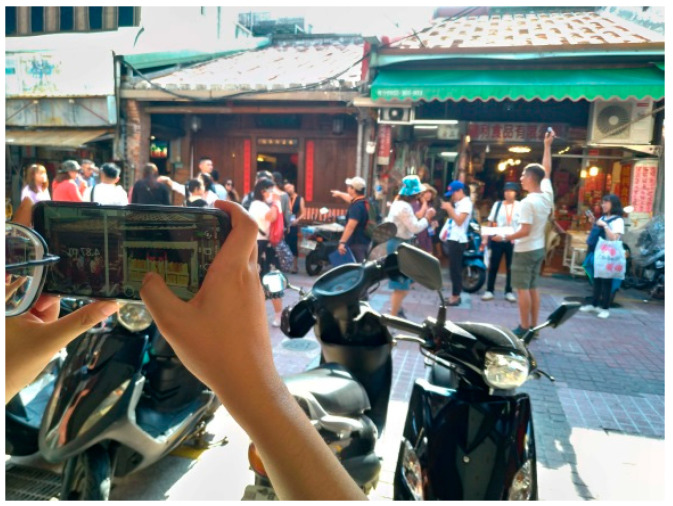
One of the factors that affected the situated comparison by interference of pedestrians or vehicles.

**Table 1 sensors-20-05628-t001:** The reviewing record of situated comparisons.

	**Adjustments & Correlations**		**Adjust 3D Model**									
			**Transformation**			**Alignment**						
**Locations**	**Address**		**Transition**	**Rotation**	**Scaling**	**Ground**	**Corner**	**Wall**	**Column**	**Roof**	**Ceiling**	
Lukang	1	44, Houche Ln. (A)	v	v	v	v						
	2	47-1, Houche Ln. (B)	v	v	v	v			v			
	3	49, Houche Ln. (C)	v	v	v	v			v			
	4	65, Houche Ln. (D)	v	v	v	v			v			
Dihua St. Taipei	5	24. Sect. 1, Dihua St. (A)	v	v	v	v		v	v		v	
	6	26. Sect. 1, Dihua St. (B)	v	v	v	v	v	v	v		v	
	7	28. Sect. 1, Dihua St. (C)	v	v	v	v	v	v	v		v	
	8	156. Sect. 1, Dihua St. (D)	v	v	v	v	v	v	v	v		
Bopiliao Taipei	9	A	v	v	v	v		v	v		v	
	10	B	v	v	v	v	v	v			v	
	11	C	v	v	v	v	v	v			v	
	12	D	v	v	v	v		v				
	13	E	v	v	v	v		v		v		
Chifeng St. Taipei	14	7, 47 Ln., Chifeng St. (A)	v	v	v	v	v	v	v		v	
	15	9, 47 Ln., Chifeng St. (B)	v	v	v	v		v		v		
	16	16, 47 Ln., Chifeng St. (C)	v	v	v	v	v	v	v		v	
	17	18, 47 Ln., Chifeng St. (D)	v	v	v	v	v	v			v	
	18	24, 47 Ln., Chifeng St. (E)	v	v	v	v	v	v			v	
	**Adjustments & Correlations**		**Correlate Details**									
			**Dimensions**				**Partitions**			**Materials**		
**Locations**	**Address**		**Width**	**Floor Height**	**Door W/H**	**Window W/H**	**Bay No.**	**Moldings**	**V/H Orientation**	**Wood**	**Brick**	**Concrete**
Lukang	1	44, Houche Ln. (A)	v	v			3	v	v	v	v	
	2	47-1, Houche Ln. (B)	v	v			3	v	v			v
	3	49, Houche Ln. (C)	v	v	v		3	v	v			v
	4	65, Houche Ln. (D)	v	v			3	v	v			v
Dihua St. Taipei	5	24. Sect. 1, Dihua St. (A)	v	v			3			v	v	v
	6	26. Sect. 1, Dihua St. (B)	v	v			3				v	v
	7	28. Sect. 1, Dihua St. (C)	v	v			3					v
	8	156. Sect. 1, Dihua St. (D)	v	v			3			v		
Bopiliao Taipei	9	A	v	v			3				v	v
	10	B		v			3			v	v	
	11	C		v			3			v	v	
	12	D	v	v			3				v	v
	13	E	v	v			3			v	v	
Chifeng St. Taipei	14	7, 47 Ln., Chifeng St. (A)	v	v			3					v
	15	9, 47 Ln., Chifeng St. (B)		v			3					v
	16	16, 47 Ln., Chifeng St. (C)	v	v			3				v	v
	17	18, 47 Ln., Chifeng St. (D)	v	v			3				v	v
	18	24, 47 Ln., Chifeng St. (E)		v	v		3	v	v	v		v
	**Adjustments & Correlations**		**Correlate Components**									
**Locations**	**Address**		**Eaves**	**Cantilevels**	**Canopies**	**Pedestrian Arcades**	**Front Steps**		**Spring Couplets**		**Openings**	
Lukang	1	44, Houche Ln. (A)	v	v	v				v			
	2	47-1, Houche Ln. (B)							v			
	3	49, Houche Ln. (C)							v		v	
	4	65, Houche Ln. (D)							v			
Dihua St. Taipei	5	24. Sect. 1, Dihua St. (A)			v	v					v	
	6	26. Sect. 1, Dihua St. (B)				v					v	
	7	28. Sect. 1, Dihua St. (C)				v					v	
	8	156. Sect. 1, Dihua St. (D)	v			v			v		v	
Bopiliao Taipei	9	A	v	v		v	v				v	
	10	B	v	v		v					v	
	11	C	v			v					v	
	12	D	v								v	
	13	E	v			v					v	
Chifeng St. Taipei	14	7, 47 Ln., Chifeng St. (A)	v	v	v	v					v	
	15	9, 47 Ln., Chifeng St. (B)		v								
	16	16, 47 Ln., Chifeng St. (C)	v	v		v					v	
	17	18, 47 Ln., Chifeng St. (D)	v	v		v					v	
	18	24, 47 Ln., Chifeng St. (E)	v	v		v					v	

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
