# Peer review of "LiDAR- and AR-Based Monitoring of Evolved Building Façades upon Zoning Conflicts"

_sensors, 2020, doi:10.3390/s20195628_

Round 1
Reviewer 1 Report
The work you have done to improve the manuscript is evident.
Well done, and an interesting article.
Just a few comments:
Careful to some English typos (i.e naviagtion, or succesfully, in the Introduction; they must be navigation, successfully: some automatic corrector would be useful).
I am not sure about reference [31], if all the long link can be included as it is shown.
Since in the Introduction you highlight: “For example, augmented reality (AR) has been widely and succesfully (note: successfully) applied in recent years. AR enables real and virtual information in an actual environment to be combined, registered, and interacted upon in real-time [1-3]. This interaction is particularly important in tacit knowledge, which is typically challenging to articulate to others [4]. Similar to case studies of environments prior to architecture design, AR can present virtual content with the real world around the user [5]”, please see if the following work may turn useful:
• Ullo, S.L., Piedimonte, P., Leccese, F., De Francesco, E., “A step toward the standardization of maintenance and training services in C4I military systems with Mixed Reality application”, Measurement: Journal of the International Measurement Confederation, 138, pp. 149-156
where advantages of Augmented and Mixed reality are highlighted when skilled people are used remotely to teach specific process steps to workers or maintainers, farly situated, in different places; and the main challenges of Augmented Reality applied to training and maintenance are discussed, related to the necessity of standardization.
Author Response
Dear Reviewer:
On behalf of my co-author, I would like to appreciate your reviewing effort. It is a time- and effort-consuming task. Through the comments you made to my manuscript, I am enlightened.
Your assistance is highly appreciated.
Best regards,
Naai-Jung Shih

Reviewer 2 Report
The revised version of "LiDAR- and AR-based Monitoring of Evolved Building Façades upon Zoning Conflicts" (Sensor #901599) has resolved the problems I raised in the last round reviewing.
One minor comment to the citation [31]. I cannot access the file source. The authors are suggested to replace with another source, or find another journal paper to support their argument.
Author Response
Dear Reviewer:
On behalf of my co-author, I would like to appreciate your reviewing effort. It is a time- and effort-consuming task. Through the comments you made to my manuscript, I am enlightened.
Your assistance is highly appreciated.
Best regards,
Naai-Jung Shih

This manuscript is a resubmission of an earlier submission. The following is a list of the peer review reports and author responses from that submission.
Round 1
Reviewer 1 Report
The manuscript “LiDAR- and AR-based Smartphone Platform for Monitoring and Mapping of Building Façade Profiles upon Zoning Conflicts” (No. sensors-901599) aims to present a technological pipeline, starting from LiDAR and ending with AR, for inspecting facades of cultural heritages. However, the limited technological innovation and unconvincing narrative hindered the quality of the paper.
A reject is recommended. However, a resubmission is welcome if the following problems can be fixed.
- A conflict between the title and content
The word “Smartphone Platform” in the title suggests that the key innovation is about a smartphone platform. However, the AR phone software—i.e., the Augment APP—in the paper was not developed by the authors. Upon my desktop search, it was developed by either Augment® (named Augment) or Univ. of National Singapore (named AUGMent) on Google Play Store.
What puzzled me more is the authors have successful experiences in developing AR phone apps such as ARTS (Shih et al. 2019a). The ARTS app was developed exactly for the same study area—Old Street—with decent AR capacity (e.g., 1M polygons). I do not understand why the authors bother to use or improve it.
Therefore, one way is to remove the “Smartphone Platform” from the title. The authors can improve their previous APP to the new application, too.
- The whole Paragraph 2 (Lines 36-42) is cloned from Shih et al. (2019b); but without a proper citation.
- Weak literature review
LiDAR, AR, and cultural heritage are all thriving research fields with exciting developments in many directions every year. A weak, or missing to some extent, literature review confuses the readers and—more critically—undermines the declarations of the contributions of the paper. This is another reason I must recommend a reject to the current version of the paper.
- Tortuous research design
Sections 4, 5, and 6 are scarcely connected in terms of research design. A reader would expect to see a holistic overview of the software platform in Sect 4, and landscape/heritage-specific functions (in current Sects. 4-6) and implementations in the subsequent section.
Sect 7: Background information on the study area should be moved to Sect 3.
The contributions, limitations, and future works are not presented in Discussion and Conclusion.
- Unconvincing experiments
There is only one screenshot from one testing site/case for validation. The small number of tests are incapacitated to convince readers of the applicability, reproducibility, and contributions of the methodology. For such an application innovation paper, a reader usually expects to see: (1) 4 or more cases/sites, (2) all the functions (as defined in the software design) on screenshots, and (3) third-person photos about site operations (e.g., Shih et al. 2019a).
- Minor issues:
- Line 79: “3.3. D scan” --> “3. 3D scan” ?
- Line 88: What model of Faro Focus 3D scanners?
References:
Shih, N. J., Diao, P. H., & Chen, Y. (2019a). ARTS, an AR Tourism System, for the Integration of 3D Scanning and Smartphone AR in Cultural Heritage Tourism and Pedagogy. Sensors, 19(17), 3725.
Shih, N. J., Hsu, W. T., & Diao, P. H. (2019b). Point Cloud-Oriented Inspection of Old Street’s Sustainable Transformation from the Ceramic Industry to Cultural Tourism: A Case Study of Yingge, a Ceramic Town in Taiwan. Sustainability, 11(17), 4749.
Reviewer 2 Report
The paper gives an interesting analysis of Old Street fabrics and architectural components in Lukang, Taiwan.
Yet, there are several issues with the paper that do not make it suitable for publication without major revisions of the same.
In the Abstract, and later in the Conclusions as well, the motivations of the work are not clear, main objectives, novelty with respect to other similar works.
Technical and theoretical assumptions are missing, for instance: at line 64, authors are talking about 3D laser scanning, which LiDAR device has been used? which is the point cloud density of the measurements? and so on.
About Augmentd Reality (AR) and LiDAR combination, the state-of-the art must be reviewed in detail.
Why do you think that your proposed methodology performs better?
Describe clearly the methodoly and what are the advantages over other methods.
I can suggest some references, but many other important ones can be found:
1) T. Pylvänäinen, J. Berclaz, T. Korah, V. Hedau, M. Aanjaneya and R. Grzeszczuk, "3D City Modeling from Street-Level Data for Augmented Reality Applications," 2012 Second International Conference on 3D Imaging, Modeling, Processing, Visualization & Transmission, Zurich, 2012, pp. 238-245, doi: 10.1109/3DIMPVT.2012.19.
2) Banfi, F. & Brumana, R. & Stanga, Chiara. (2019). Extended reality and informative models for the architectural heritage: from scan-to-BIM process to virtual and augmented reality. Virtual Archaeology Review. 10. 14. 10.4995/var.2019.11923.
3) Ullo, S.L.; Zarro, C.; Wojtowicz, K.; Meoli, G.; Focareta, M. LiDAR-Based System and Optical VHR Data for Building Detection and Mapping. Sensors 2020, 20, 1285.
Besides that, I recommend a strong revision of English, and attention to typos (i.e. navigation at 39 line).
Number of keywords must be increased.
All acronyms must be written in full, and identified in the paper the first time they appear (i.e. AR, etc.)